# Fault Diagnosis Method for Rolling Mill Multi Row Bearings Based on AMVMD-MC1DCNN under Unbalanced Dataset

**DOI:** 10.3390/s21165494

**Published:** 2021-08-15

**Authors:** Chen Zhao, Jianliang Sun, Shuilin Lin, Yan Peng

**Affiliations:** National Cold Rolling Strip Equipment and Process Engineering Technology Research Center, Yanshan University, Qinhuangdao 066000, China; zhaochen136@stumail.ysu.edu.cn (C.Z.); lslin@stumail.ysu.edu.cn (S.L.); pengyan@ysu.edu.cn (Y.P.)

**Keywords:** Adaptive Multivariate Variational Mode Decomposition, Multi-channel One-Dimensional Convolutional Neural Network, deep convolutional generation adversarial network, unbalanced dataset fault diagnosis, rolling mill multi-row bearings

## Abstract

Rolling mill multi-row bearings are subjected to axial loads, which cause damage of rolling elements and cages, so the axial vibration signal contains rich fault character information. The vertical shock caused by the failure is weakened because multiple rows of bearings are subjected to radial forces together. Considering the special characters of rolling mill bearing vibration signals, a fault diagnosis method combining Adaptive Multivariate Variational Mode Decomposition (AMVMD) and Multi-channel One-dimensional Convolution Neural Network (MC1DCNN) is proposed to improve the diagnosis accuracy. Additionally, Deep Convolutional Generative Adversarial Network (DCGAN) is embedded in models to solve the problem of fault data scarcity. DCGAN is used to generate AMVMD reconstruction data to supplement the unbalanced dataset, and the MC1DCNN model is trained by the dataset to diagnose the real data. The proposed method is compared with a variety of diagnostic models, and the experimental results show that the method can effectively improve the diagnosis accuracy of rolling mill multi-row bearing under unbalanced dataset conditions. It is an important guide to the current problem of insufficient data and low diagnosis accuracy faced in the fault diagnosis of multi-row bearings such as rolling mills.

## 1. Introduction

Rolling mill multi-row bearings are the core of the main drive system of the rolling mill, which support the rolling mill roll system and withstand huge radial forces. Under the working conditions, the rolls have axial displacement and roll bending phenomenon, the bearing can absorb the harmful bending moment and axial force. Therefore, the cage and rolling body are the main failure parts of the rolling mill multi-row bearing. According to statistics, 30% of rotating machinery failures are caused by bearings, and their operating conditions directly affect system performance [1,2]. If the multi-row bearings of large machinery such as the rolling mill are damaged, it can lead to long downtime, and extremely high repair costs and serious economic losses. Therefore, it is necessary to carry out the diagnosis of bearing faults in large machinery and equipment such as rolling mills. Due to the harsh factory conditions and noise interference, the vibration signal has nonlinear and non-stationary characters [3]. Therefore, conventional time domain waveform and frequency domain feature fault analysis methods have limitations in bearing fault diagnosis [4].

The time-frequency analysis method has better results in dealing with nonlinear and non-stationary signals, which has been widely used in fault diagnosis. The following methods are commonly used: Empirical Mode Decomposition (EMD) [5], Local Mean Decomposition (LMD) [6], Empirical Wavelet Transform (EWT) [7], Variational Mode Decomposition (VMD) [8], etc. Among them, VMD changes the previous signal processing and decomposes the signal according to the center frequency, which makes the characters of Intrinsic Mode Function (IMF) much more controllable. In [9], Li compared the effectiveness of VMD and EMD in processing vibration signals, which proved that VMD outperforms EMD and can effectively overcome the problem of modal mixing. In [10], Aneesh considered the classification of power quality disturbances based on VMD and EWT, and classification results indicated that VMD outperformed EWT for feature extraction. However, the above algorithms have limitations in processing multidimensional signals, so in [11], Rehman proposed Multivariate Empirical Mode Decomposition (MEMD). Based on the idea of MEMD, in [12], Aftab and Rehman extended VMD to multidimensional and proposed Multivariate Variational Mode Decomposition (MVMD), which effectively solves the problem of synchronous processing of multivariate data. The literature [13] showed that the effect of VMD decomposition was greatly influenced by the parameters K and α, and it cannot achieve adaptive decomposition of the signal in a real sense. Therefore, MVMD as an extension of VMD also has a parameter optimization problem. Although the multiple signal input activates the noise reduction capability of the Wiener filter and reduces the effect of the number of IMF K on the decomposition effect [14], the iterative optimization-seeking process of MVMD converges too slowly, and the decomposition effect is still affected by the penalty factor α.

In view of the superior performance of mode decomposition, scholars have combined it with pattern recognition methods to become the mainstream fault diagnosis method. In [15], Isham used VMD to reconstruct wind turbine gearbox vibration signals and extracted multi-domain features that were passed to an Extreme Value Learning Machine (ELM) for fault classification. The ELM requires fewer samples for training and has a fast speed on diagnosis, but the relative stability of the model is weaker [16]. In [17], Gu used MVMD to decompose diesel multi-sensor signals for processing, but still needed to use band entropy for feature extraction in the process of combining Support Vector Machines (SVM). However, the kernel function selection of SVM has a large impact on the classification, and the classification effect is significantly affected by the fault samples. Therefore, SVM often needs to be combined with optimization algorithms, which increases the tediousness of the model [18]. Because conventional classifiers such as ELM and SVM need to be combined with feature extraction methods, the fault diagnosis method deviates from the general trend of end-to-end (signal-to-fault) diagnosis.

Convolutional Neural Networks (CNNs) have had significant achievements in the field of image recognition and have become a research hotspot in deep learning [19]. CNN has the function of automatic feature extraction and pattern recognition, which can realize the fault identification of equipment by inputting vibration signal. Therefore, CNN is widely used in end-to-end fault diagnosis. There are two main modes of application of CNN in fault diagnosis. On the one hand, the vibration data is transformed into a two-dimensional data matrix for identification. In [20], Chen transformed a certain length of one-dimensional vibration signal into a two-dimensional matrix and used CNN for fault identification. In [21], Xu used the IMF component signal of VMD as the input of CNN and achieved good results in the fault diagnosis of wind turbine bearings. On the other hand, vibration data can be transformed into image formats such as grayscale images, frequency domain maps and speech spectrum maps for recognition. In [22], Zhu transformed the signal by short-time Fourier transform into a frequency domain map for fault diagnosis by CNN. In [23], Zhao transformed the one-dimensional vibration signal into a two-dimensional grayscale image and achieved diagnostic classification of faults by CNN. However, the vibration signal is a one-dimensional time series signal, and the data at each moment have a certain correlation. Converting one-dimensional data into two-dimensional arrays and performing feature extraction by convolutional kernels can break the spatial correlation of signals, resulting in the loss of fault character information. Therefore, scholars have proposed One-Dimensional Convolutional Neural Network (1DCNN) for the special characteristics of one-dimensional time series. In [24], Levent directly input the raw vibration signal of the bearing into 1DCNN to achieve rapid diagnosis of bearing faults. In [25], Wu used 1DCNN for the fault diagnosis study of gearboxes, which reflected the strong feature extraction and recognition classification ability of 1DCNN. One-dimensional convolution solves the problem of time series feature loss, but also makes CNN lose the ability to handle high-dimensional data; the analysis of a single-channel signal cannot fully explore the fault character information of the large equipment. Moreover, the actual signal of the engineering contains a large number of invalid character components and noise, which greatly reduces the feature extraction ability of 1DCNN.

The powerful classification ability of CNN also requires a large amount of data for training. However, in order to ensure production safety, fault equipment needs to be shut down in time, which makes it difficult to obtain a large amount of fault data, and the model is poorly trained. In [26,27], GAN and its variants had been shown to generate audio data and EEG signal which showed their potential to generate time-series data. In [28], Liu applied Generating Adversarial Network (GAN) to deep feature enhancement of bearing data and demonstrated that GAN can overcome the problems of insufficient fault data and unbalanced dataset, and GAN can improve the model training effect to improve the diagnosis accuracy. However, the basic GAN model suffers from gradient disappearance, pattern collapse, poorer results generated by the generator, and growth in the training time of the model [29]. In [30], Radford built the GAN layer structure by convolution and deconvolution to form the DCGAN algorithm, which greatly improves the performance of GAN. In [31,32], Guo and Gao both used 1DCNN to construct the layer structure of GAN and achieved better results in bearing fault diagnosis under the condition of an unbalanced dataset. Although DCGAN largely solves the problems of poor generation results and the long training time of GAN, the presence of large noise interference in the original signal and invalid feature information still leads to the limitations of DCGAN in dataset enhancement.

Based on the existing work, we considered the unique fault characteristic distribution of axial and vertical vibration signals of multi-row rolling bearings in rolling mill and the problem of an unbalanced dataset in practical applications, so we introduced a multi-channel signal fault diagnosis method of unbalanced datasets to the field of similar bearing fault diagnosis. In this paper, MVMD is used to process multi-channel signals, but the effect of both VMD and MVMD is greatly influenced by the parameters K and α [17,33]. Therefore, we proposed an Adaptive Multivariate Variational Mode Decomposition (AMVMD) signal processing method. Using the mean of Weighted Permutation Entropy (WPE) as the fitness factor, we used the Genetic Algorithm (GA) to implement the optimal selection of parameters K and α and introduced an iterative operator to accelerate the iterative merit seeking of MVMD. Because of the limitations of 1CDNN in processing multi-channel signals, we proposed Multi-channel One-Dimensional Convolutional Neural Network (MC1DCNN) by introducing the multichannel convolutional fusion layer at 1DCNN, which makes up for the shortcomings of 1DCNN in multi-channel signal processing. In order to reduce the effect of noise on the feature extraction ability of MC1DCNN, AMVMD was combined with MC1DCNN and applied to multi-channel signal fault diagnosis of rolling mill multi-row bearings. Considering the problem that fault data is difficult to obtain and the networks could not achieve good diagnostic accuracy under the condition of unbalanced dataset [34], a Deep Convolutional Generative Adversarial Network (DCGAN) was embedded in the model training process. Additionally, thanks to the excellent signal processing effect of AMVMD, it can effectively reduce the invalid feature information and noise interference in the signal and improve the dataset enhancement capability of DCGAN. Finally, we realized the construction of a fault diagnosis model under an unbalanced dataset. 

The rest of the paper is organized as follows: Section 2 describes the optimization algorithm (GA and Iterative acceleration operator) and optimization process of AMVMD proposed in this paper and describes the theory and network structure of DCGAN. In Section 3, the simulated signal is used for analysis in order to better represent the data enhancement effect of the method in Section 2. Section 4 describes the theory of MC1DCNN, combines it with AMVMD and embeds the DCGAN module in the model to form a fault diagnosis model under an unbalanced dataset. Section 5 applies the model of this paper to the fault diagnosis of the rolling mill fault simulation test bench and gets good results. Additionally, we compare this model with the approximate model and existing models to prove the advantage of this model. Finally, the conclusion is drawn in Section 6.

## 2. AMVMD Signal Processing and Unbalanced Data Generation

### 2.1. Iterative Acceleration of MVMD

The MVMD algorithm has been recently proposed to solve the problem of cooperative decomposition of multi-channel data and to solve the problem that VMD can only handle single-channel signal. The multi-channel signal can excite the noise reduction ability of the Wiener filter and improve the signal processing effect of MVMD. MVMD converts the IMF component of the multi-channel signal into a set of AM-FM signals as *u*(*t*):(1)$$u(t)=u_{c}(t)=a_{c}(t)\cos(\phi_{c}(t))$$
where *a**_c_*(*t*) is the amplitude of the *c*-th component and *φ**_c_*(*t*) is the phase of the *c*-th component.

Taking the square of the *L*^2^ parametric of the mixed signal to find the *u*(*t*) bandwidth, and then the constrained variational optimization of the *u*(*t*) bandwidth of the multi-channel signal is performed. It is required to minimize the bandwidth sum of the individual components separated in the c signals, while ensuring the accuracy of each classification, modeled as follows.
(2)$$\{\begin{array}{c} \underset{\left\{u_{K,c}\}\{\omega_{k}\right\}}{\min}\{{\sum_{K}\sum_{c}\left\|\partial_{t}[u_{+}^{K,c}(t)e^{-j\omega_{K}t}]\right\|}_{2}^{2}\},\\ \sum_{K}u_{K,c}(t)=x_{c}(t),c=1,2,3,\cdots ,c \end{array}$$
where *K* is the number of IMF, *c* is the number of channels of the input signal, and *ω_k_* is the center frequency of each mode.

The constrained variational model is constructed by using the Lagrange multiplier method and is transformed into an unconstrained variational problem by introducing the penalty factor *α* with the Lagrange multiplier *λ*(*t*). Construct the Lagrange function model as follows:(3)$$\begin{array}{l} L(\{u_{K,c}\},\{\omega_{K}\},\lambda_{c})=\alpha \sum_{K}\sum_{c}\left\|\partial_{t}[u_{+}^{K,c}(t)e^{-j\omega_{K}t}]\right\|\\ +{\sum_{c}\left\|x_{c}(t)-\sum_{K}u_{K,c}(t)\right\|}_{2}^{2}+\sum_{c}\langle \lambda_{c}(t),x_{c}(t)-\sum_{K}u_{K,c}(t)\rangle \end{array}$$

The alternating direction multiplier method is used to transform the optimization problem into a suboptimization problem, and the optimal mode and center frequency of the multivariate signal are obtained by iteratively updating the subproblem.

In this paper, to address the problem of the slow iterative search speed of MVMD, an iterative operator is introduced to accelerate the solution process, and the specific iterative process is as follows.(1)Initialize $\left\{{\hat{u}}_{K,c}^{1}\right\}$,$\left\{\omega_{K}^{1}\right\}$,${\hat{\lambda }}_{c}^{1}$, set n=0, $\varepsilon {=10}^{-7}$.(2)Set n=n+1, and execute a loop to update $\left\{{\hat{u}}_{K,c}^{n+1}\right\}$,$\left\{\omega_{K}^{n+1}\right\}$ and ${\hat{\lambda }}_{c}^{n+1}$ until iterative precision is reached.
(4)$${\hat{u}}_{K,c}^{n+1}(\omega )=\frac{{\hat{x}}_{c}-\sum_{i\neq K}{\hat{u}}_{i,c}(\omega )+\frac{{\hat{\lambda }}_{c}(\omega )}{2}}{1+2\alpha {\left(\omega -\omega_{K}\right)}^{2}}$$
(5)$$\omega_{K}^{n+1}=\frac{{\sum_{c}\int_{0}^{\infty }\omega \left|{\hat{u}}_{K,c}(\omega )\right|}^{2}d\omega }{{\sum_{c}\int_{0}^{\infty }\left|{\hat{u}}_{K,c}(\omega )\right|}^{2}d\omega }$$Update ${\hat{\lambda }}_{c}^{n+1}$ for all *ω >* 0
(6)$$t_{n+1}=(1+\sqrt{1+4t_{n}{}^{2}})/2$$
(7)$${\hat{\lambda }}_{c}^{n+1}(\omega )={\hat{\lambda }}_{c}^{n}(\omega )+\tau ({\hat{x}}_{c}(\omega )-\sum_{K}{\hat{u}}_{K,c}^{n+1}(\omega ))$$
(8)$${\hat{\lambda }}_{n+1}(\omega )={\hat{\lambda }}_{n+1}(\omega )+(\frac{t_{n}-1}{t_{n+1}})[{\hat{\lambda }}_{n+1}(\omega )-{\hat{\lambda }}_{n}(\omega )]$$(3)Stop the iteration when the iteration accuracy is satisfied and output the set of modes *u_K_* and the center frequency *ω_K_*.
(9)$$\sum_{K}\sum_{c}\frac{{\left\|{\hat{u}}_{K,c}^{n+1}-{\hat{u}}_{K,c}^{n}\right\|}_{2}^{2}}{{\left\|{\hat{u}}_{K,c}^{n}\right\|}_{2}^{2}}<\varepsilon$$

### 2.2. Parameter Optimization Based on GA

GA is able to search for the optimal solution in a complex space. The WPE can reflect the randomness and complexity of the time series, and the smaller WPE proves that the signal is more regular and contains more information of fault characteristics. Calculate the average WPE of multi-channel IMF to evaluate the decomposition effect and use it as the fitness function of GA. The parameters to be optimally selected are *K* and α. Therefore, each chromosome of GA is coded as {*Xi*:* K, α*}. So, the fitness of GA is calculated as follows.
(10)$$F=\min(\frac{1}{K}\sum_{i}^{K}\mathrm{WPE}({\mathrm{IMF}}_{i},m,\tau ))$$
where *m* is the embedding dimension, which is set to 4; *τ* is the delay time, which is set to 1; and *K* is the number of IMFs.

Individuals with better fitness are selected as parents of the next generation, and *X*_1_ and *X*_2_ are randomly selected from these chromosomes for crossover to obtain new offspring $X_{1}^{\prime }$ and $X_{2}^{\prime }$.
(11)$$\begin{array}{l} X_{1}^{\prime }=\lambda X_{1}+(1-\lambda )X_{2}\\ X_{2}^{\prime }=\lambda X_{2}+(1-\lambda )X_{1} \end{array}$$
where *λ* is the crossover factor, *λ*∈[0,1].

Select a chromosome *X* randomly and select a gene *i* from chromosome *X*, then mutate gene *i* to obtain its mutation value *U* (*i*_min_, *i*_max_). The optimal combination of parameters [*K*_best_, *α*_best_] is finally obtained after the optimization iteration of GA.

### 2.3. Unbalanced Data Generation Based on DCGAN

GAN is proposed inspired by game theory and consists of a generator *G* and a discriminator *D*. Through training, the generator keeps learning and the discriminator keeps becoming optimized [35]. Input random noise *z* into *G* for data generation, and the model expects the generated data *G*(*z*) to be discriminated as true by *D*, i.e., *D*(*G*(*z*)) = 1. For the discriminator *D*, it is expected that when the input is *G*(*z*), *D* discriminates it as false, i.e., as *D*(*G*(*z*)) = 0. That is, for the problem of minimizing *G* and maximizing *D*, the discriminator and generator model loss functions are shown in (12) and (13).
(12)$$\underset{D}{\max}V(D,G)=E_{x-P_{data(x)}}[\log(D(x))]+E_{z-P_{g}(z)}[\log(1-D(G(z)))]$$
(13)$$\underset{G}{\min}V(D,G)=E_{z-P_{g}(z)}[\log(1-D(G(z)))]$$

Through adversarial learning, the functions of *G* and *D* are continuously improved, and the final mathematical model is as follows.
(14)$$\underset{G}{\min}\;\underset{D}{\max}V(D,G)=E_{x-P_{data}(x)}[\log D(x)]+E_{z-P_{g}(z)}[\log(1-D(G(z)))]$$
where *x* is the real sample; *P_data_* is the distribution of real data; and *P_g_* is the distribution of noise.

Radford proposed a DCGAN algorithm, which greatly improves the performance of GAN [30]. Additionally, in [36], Mirza restricted the generation process by inputting conditional variables to solve the problem that the training process is unstable with the generation results and the generated samples differ from the generation target, which in turn guides the generation of the desired samples. In this paper, DCGAN is used to generate the AMVMD reconstructed signal, and the generator mainly consists of four deconvolutional layers and the discriminator mainly consists of four convolutional layers, as shown in Figure 1. The reconstructed signal removes the invalid features and retains the faulty features, which can reduce the generation of invalid features by DCGAN and improve the ability of DCGAN to generate virtual samples.

## 3. Analysis of Simulated Signals

### 3.1. Construction of Simulation Signal

The rolling mill multi-row rolling bearing vibration signal is a non-linear, non-stationary modulated signal; according to the actual working conditions, we set the amplitude modulation signal (*x*_1_), frequency modulation signal (*x*_2_), and harmonic signal (*x*_3_) to simulate vibration signal. Each frequency of the simulated signals are as follows: *f*_1_ = 80 Hz, *f*_2_ = 30 Hz, *f*_3_ = 200 Hz, *f*_4_ = 50 Hz, *f*_5_ = 300 Hz, and the main characteristic frequencies are *f*_1_, *f*_3_ and *f*_5_.
(15)$$\left\{ \begin{array}{l} x_{1}=\cos(2\pi f_{1}t)[1+\sin(2\pi f_{2}t)]\\ x_{2}=\sin[2\pi f_{3}t+\cos(2\pi f_{4}t)]\\ x_{3}=\sin(2\pi f_{5}t) \end{array}\right.$$

In the actual signal acquisition, due to the complex transmission path and noise interference, the sensor acquisition vibration signal is different, so we simulate each channel signal with different weighting ratios for the three simulated signals as follows.
(16)$$\left\{ \begin{array}{l} s_{1}=0.45x_{1}+0.85x_{2}+0.62x_{3}+n_{1}\\ s_{2}=0.85x_{1}+0.7x_{2}+0.35x_{3}+n_{2}\\ s_{3}=0.6x_{1}+0.4x_{2}+0.9x_{3}+n_{3} \end{array}\right.$$
where *n*_1_, *n*_2_, *n*_3_ and are 25 db, 18 db and 13 db noise signals, respectively.

The time domain waveforms and frequency domain character of the three-channel simulated signal are shown in Figure 2.

### 3.2. Algorithm Performance Comparison

We use MEMD, MVMD and AMVMD to decompose the simulated signal, set the number of modes *K* of MVMD to 4 and the penalty factor *α* to 2000, and set the K and α of AMVMD to 4 and 2434 after optimization by GA, respectively. The time required for MVMD and AMVMD to process signals of different lengths was calculated 10 times and averaged, and the results are shown in Table 1. The operating environment is Windows 10, the CPU is Intel i7-9750H (2.60 GHz), and the RAM is 16 GB. The AMVMD computation time is significantly reduced after the introduction of the iterative operator.

Fourteen groups of IMFs are obtained by MEMD, and we only take the first five groups of IMFs for frequency domain analysis, as shown in Figure 3a, the MVMD decomposition results as shown in Figure 3b, and the AMVMD decomposition results as shown in Figure 3c. It can be seen from Figure 3 that all three algorithms adaptively decompose the multivariate simulation signal to obtain the IMF component of the response principal frequency. However, some of the same frequency components are reflected in different IMFs, i.e., the phenomenon of mode mixing appears. The most serious modal mixing is found in the IMFs of MEMD, where IMF3 has a primary frequency of 150 Hz (*f*_3_ − *f*_4_) and IMF5 has a primary frequency of 50 Hz (*f*_2_ − *f*_1_); both frequencies are the sideband frequencies of the primary frequency peak of the original signal. Additionally, there are more cluttered noise frequencies in the IMFs of MEMD, while the IMFs of MVMD and AMVMD are basically free of noise frequencies.

In Figure 3b,c, the IMFs are well decomposed according to the major center frequencies, and the corresponding side bands appear on both sides of the major center frequencies, and the side band frequencies in IMF1 are 50 Hz and 110 Hz (*f*_1_ ± *f*_2_), and the side band frequencies in IMF2 are 150 Hz and 250 Hz (*f*_3_ ± *f*_4_), and there is basically no main frequency peak in IMF4, and the signal is well decomposed. However, a left side band frequency of 250 Hz (*f*_3_ + *f*_4_) appears in IMF3, and the overall peak of the side band frequency of the modal mixing in IMF3 of AMVMD is reduced compared to that of MVMD.

### 3.3. Generation of Simulation Data 

We use DCGAN to generate the IMFs of the simulated signal; the training set is the IMF1–IMF3 of the three-channel simulated signal, and the sample length is set to 1024. The time-domain waveform comparison and frequency-domain character comparison of the generated signal and the real signal for each channel are shown in Figure 4. The generated signal well simulates the time-domain waveform characters and frequency-domain characters of the real signal, which can realize the supplement of scarce data.

## 4. Fault Diagnosis Model Based on AMVMD-MC1DCNN

### 4.1. One-Dimension Convolutional Neural Network 

CNN was originally applied to image recognition techniques [37]. The local connectivity, weight sharing, and down-sampling character of CNN make the network structure massively reduced, and CNN can make full use of the local features of the data itself and thus improve the computational efficiency. The structure of CNN includes convolutional layers, a pooling layer, a fully connected layer and an output layer [38]. The main difference between 1DCNN and CNN is that the input dimension of the character is one-dimensional, so 1DCNN consists of one-dimensional convolutional layers, one-dimensional pooling layers, a fully connected layer and a Softmax classifier, and the structure of 1DCNN is shown in Figure 5.

Assuming that a one-dimensional signal *x_i_* is the output of layer *i*, its convolution is computed in the following way.
(17)$$x_{j}^{l}=f(\sum_{i\in N_{j}}x_{i}^{l-1}*w_{ij}^{l}+b_{j}^{l})$$
where *N_j_* is the *j*-th convolutional region of the *l*-1st layer; $x_{j}^{l}$ is the *j*-th input to the convolution of *l* layer; *w* is the weight matrix (convolution kernel); *b* is the bias of the convolution layer; *f* is the nonlinear activation function.

The one-dimensional pooling layer, also known as the down-sampling layer, reduces the dimensionality of the convolutional features and reduces the computational effort of the classifier. The maximum pooling process is usually chosen to ensure the invariance of feature scales and reduce the size of the input data.
(18)$$x_{j}^{l}=f(down(x_{j}^{l-1}+b_{j}^{l}))$$
where *down*() is sampling function. 

The fully connected layer can rearrange the characters extracted from the previous convolutional layers and pooling layers into a column, and the Dropout function is usually added to suppress overfitting and improve generalization ability of CNN.
(19)$$\delta i=f(w_{i}p_{i}+b_{i})$$
where i=1,2,⋯,k, δi is the *i*-th output, and *k* in total.

The most commonly used classifier for CNN is the supervised learning Softmax classifier. Additionally, the network is optimally trained using the Adam optimization algorithm, which in turn accomplishes the multi-classification task. The output of Softmax can be viewed as a probability problem.
(20)$$p(i)=\frac{e^{\delta_{i}}}{\sum_{k=1}^{K}e^{\delta_{k}}}$$
where *p*(*i*) is the probability of each output, the sum of *p*(*i*) is 1, and *K* is the number of categories. 

### 4.2. Multi-Channel One-Dimension Convolutional Neural Network

In this paper, a multi-channel one-dimensional convolutional fusion layer is added to 1DCNN, as shown in Figure 6. M1DCNN can be used for multi-channel signal processing, which can synthetically consider multiple directional vibration signals for fault diagnosis analysis, and AMVMD reconstructed signal can further reduce noise interference and highlight fault characters by multi-channel one-dimensional convolution processing.

When the input to the convolution layer is a multi-channel signal, a multi-channel convolution kernel is used for the operation, and a one-dimensional convolution operation is performed in each channel individually. In order to add the correlation of the respective channels, we need to compute the weighted summation of each channel at the same position to obtain the 1D convolutional output at that position.
(21)$$x^{l}=\sum_{j=1}^{m}f_{j}(\sum_{i=1}^{k}(x_{ij}^{l-1}\times w_{ij}^{l-1})+b_{ij}^{l-1})$$
where $x^{l}$ is the output of the *l*-th convolutional layer; $x_{ij}^{l-1}$ is the *i*-th character input of the *l*-1st convolutional layer of channel *j*, with *k* character inputs; $w_{ij}^{l-1}$ is the *i*-th convolutional kernel of the *l*-1st layer of channel *j*; $b_{ij}^{l-1}$ is the ith bias value of the *l*-1st layer of channel *j*; *f* is the nonlinear activation function; *m* is the number of channels. 

The multi-channel one-dimensional convolutional fusion layer can effectively realize the fusion and character extraction of multi-channel signals. The pooling layer of M1DCNN also uses maximum pooling, followed by a fully connected layer and a Softmax classifier.

### 4.3. Fault Diagnosis Model

The fault diagnosis model based on AMVMD-M1DCNN proposed in this paper is shown in Figure 7, which consists of a multichannel input layer, multivariate mode variational reconstruction, conditional deep convolutional generation adversarial network, 1D convolutional, 1D pooling, and fully connected and Softmax classifiers.

Considering the rich fault information in the axial vibration signal of the rolling mill multi-row bearings, and the low signal-to-noise ratio of the vertical vibration signals, the axial and vertical vibration signals are input into the fault diagnosis model simultaneously. The input signal is a two-channel one-dimensional signal of 1 × 2048 × 2. The model consists of two parts: offline training and online detection. The existing label data is used to train the fault diagnosis model, and the model is used to classify the collected data. Embedding DCGAN can improve the diagnosis accuracy of fault diagnosis model under the condition of unbalanced training datasets. After each channel signal is reconstructed by AMVMD, it is input into M1DCNN for individual convolution calculation, and the multi-channel can more comprehensively explore the information of fault vibration signal characters than the single channel.

## 5. Experiments and Results Analysis

To verify the effectiveness of the diagnostic model, an experimental rolling mill fault simulation test bench was used for bearing vibration signal acquisition, and the test bench is shown in Figure 8. The parameters of the rolling mill are as follows: the diameter of the roll is 120 mm, the length of the roll is 90 mm, the speed of the main motor is 180 r/min, the maximum rolling force is 12 tons; the vibration sensor is YS8202, the acceleration sensor, and pressure sensor model is HZC-01, and the sampling frequency is 2000 Hz. The experimental bearings are double-row cylindrical roller bearings, and the bearing type is NU1012.

We collected vibration data from the operating side of the rolling mill and selected bearings with rolling element scratches, bearings with broken cages, bearings with rolling element flaking, bearings with mixed faults (rolling element flaking and broken cage) and normal bearings for fault data collection; the labels of the five types of bearings were set to 1–5 in order. The bearing failure is shown in Figure 9. In each experiment, we performed two passes of the rolling process, and we collected 120,000 data points each time, for a total of 480,000 data points in two experiments. The data are divided according to a sample length of 2048, with 240 samples available for each bearing.

### 5.1. Signal Processing by AMVMD

We used the AMVMD to decompose vertical vibration signals and axial vibration signals and then chose the better IMFs to reconstruct the signal. AMVMD had reduced the effect of the number of IMFs K on the decomposition effect; if K is set too large, the effective fault characteristics will be stripped to the worse IMFs, so we set K ∈ [3,6] and α ∈ [500,4000] and used the GA to find the optimal decomposition parameters in this range. The parameters of GA were set as follows: the population size is 10, the number of population evolution is 25, the probability of crossover is 0.8 and the probability of variation is 0.1. We recorded the best fitness and the fitness average of all individuals for each evolution. The iterative search curves for the four fault signals are shown in Figure 10, and the decomposition parameters are shown in Table 2.

The decomposition results of bearings with rolling body flaking are shown in Figure 11; the periodic waveform can be initially seen from the time domain of IMF1, IMF2 and IMF3, while the time domain waveforms of IMF4 and IMF5 are more chaotic. The frequency domain feature maps of the five IMFs appear with slight modal mixing, but the central frequencies of the individual IMFs are well separated.

We used the WPE as the evaluation index; the WPE of each IMF for eight kinds of signals are shown in Figure 12. It can be found that the WPE of the first two IMFs are significantly smaller than the other IMFs, which coincides with the regularity of the time domain waveform, and the signal period regularity is the strongest. It can be considered that IMF1 and IMF2 contain rich fault character information, so IMF1 and IMF2 are selected to reconstruct the axial vibration signal and vertical vibration signal.

### 5.2. Generate Reconstructed Data by DCGAN

The sample data length of the faulty signal and normal signal was set to 2048, and the training set of DCGAN was composed according to the unbalanced ratio of 1/10 (200 sets of normal bearing data samples and 20 sets of each fault samples), and we input the condition variables at the same time to generate the faulty bearing reconstruction signal under the unbalanced condition. The time domain waveforms and frequency domain characters of the real and generated signals are shown in Figure 13. It can be seen that the time domain waveforms of the real signal and the generated signal are similar, and the main frequency characteristics of the frequency domain maps are basically the same, and we can consider that the reconstructed signal generated by DCGAN has better fault characteristics. We combined the generated data as supplementary samples with real samples into a balanced dataset to train the fault diagnosis model and improve the accuracy of the diagnosis model under unbalanced sample conditions.

### 5.3. Fault Diagnosis by MC1DCNN

The network structure of MC1DCNN is shown in Table 3. In the first layer, we used a wide convolutional kernel to further filter out the interference of noise and save the calculation time, and the subsequent convolutional kernels use smaller convolutional kernels to fully explore the fault characters of the vibration signal.

All convolutional layers are edge-processed with the SAME function. Each convolution layer is followed by a pooling layer, and we used maximum pooling with a pooling strip width of 2. The last pooling layer is connected to the fully connected layer, which has 1024 neurons. In order to suppress the overfitting phenomenon and improve the generalization ability of the model, we added the dropout function; finally, the Softmax classifier is used for classification. 

We performed fault diagnosis analysis on the training set under three unbalanced ratios (1/20, 1/10, 1/5). We randomly selected 40 sets of various bearing data as the test set, and the remaining 200 sets of normal bearing data and the corresponding proportional quantities (10, 20, and 40) of four types of faulty bearing data as the training set. Additionally, the confusion matrix of the diagnostic results of the model under the three ratios is shown in Figure 14a. Under the condition of a lack of fault training data, the diagnosis accuracy of the model is low, and with the increased ratio of fault data to normal data, the diagnosis accuracy of the model improves.

We trained DCGAN using three unbalanced datasets and supplemented the unbalanced dataset with the data generated by DCGAN. Finally, MC1DCNN was trained with the supplemented dataset, and we obtained three balanced ratio data diagnosis models, and used the three models to identify and classify the test sets. The confusion matrix of the classification results of the three models is shown in Figure 14b. After embedding the DCGAN data supplementation module in the fault diagnosis model, the network diagnosis accuracy under the three unbalanced data conditions was significantly improved, and the diagnosis accuracy reaches more than 90% in all cases. DCGAN can effectively improve the fault diagnosis capability of the model in this paper under unbalanced data conditions.

The diagnostic results for the original signal (OS) input, the MEMD reconstructed signal input, the AMVMD reconstructed signal input and the combination of the three inputs with DCGAN are shown in Figure 15. AMVMD has the highest diagnosis accuracy under all types of ratio training sets, and when the unbalance ratio is 1/5, the model diagnosis accuracy is almost the same as the balanced data after combining AMVMD with DCGAN, which further verifies the superiority of the fault diagnosis model proposed in this paper.

### 5.4. Comparison Experiments

To further verify the superiority of the diagnosis models in this paper, we combined three processed signals (original signal, MEMD reconstructed signal and AMVMD reconstructed signal) with three classification algorithms (DBN, 1DCNN, MC1DCNN). We randomly selected 200 sets of data from each type of bearing as the training sets and the remaining 40 sets of data as the test sets and calculated the average accuracy of the model for 10 diagnoses as shown in Table 4. DBN, 1DCNN selected three modes of input (single channel vertical signal input, single channel axial signal input and dual channel signal mixing input). The implied layer number of DBN was set to 3. 1DCNN layer structure is the same as MC1DCNN as far as possible. From Table 4, it can be seen that the AMVMD-MC1DCNN model proposed in this paper has the highest diagnosis accuracy.

In order to verify the advantages of the AMVMD-MC1DCNN fault diagnosis model in this paper compared with the existing models, the existing approximate models were selected to reproduce the results for comparison. A more comprehensive comparison between 1DCNN and MC1DCNN has been made in the results of this paper, so the 1DCNN model of the literature [25,26] is not compared subsequently. The models selected for comparison are the VMD-ELM model of the literature [15], the MVMD-SVM model of the literature [17], and the VMD-CNN model of the literature [22]. Since both the VMD-ELM model and the MVMD-SVM model require feature extraction of the vibration signal, in the process of performing MWPE feature extraction, we found that the larger embedding dimension and scale factor of the Multiscale Weighted Permutation Entropy (MWPE) algorithm increase the computing time significantly, so the CNN model has an absolute advantage in diagnostic time after training is completed. Therefore, we ignore the time required for feature extraction and compare only the diagnostic accuracy of the models. Additionally, all of the above models are applied to the vibration data analysis of the experimental rolling mill bearing fault diagnosis test bench. 

The parameters of MVMD are the same as those obtained in this paper, and the parameters of VMD are also optimally selected using GA. Since the decomposition effect of SVM is affected by the kernel function parameters and penalty parameters, we used the most widely used PSO to optimize its parameters, and the kernel function of SVM was chosen as Gaussian kernel function. The PSO parameters were set as follows: the number of particles is 25, the number of iterations is 50, the local learning factor is 1.6, the global learning factor is 1.6, and the inertia factor is 0.8. The CNN network structure is designed as follows: the number of input samples is modified to 44 × 44, the number of convolutional layers is set to 4, the convolutional layer is followed by the pooling layer, the convolutional kernel size is 3 × 3, and the step size of the convolutional kernel is 1. The input of the VMD-ELM model is the multidomain features of the VMD reconstructed signal. Since the entropy algorithms can all respond to the complexity of the signal sequence, band entropy in the literature [17] was replaced with MWPE, and the entropy values of the first 20 scales were taken to construct the feature vector as the input of the SVM.

The accuracy (average of 10 diagnoses) of various existing models for the analysis of the experimental rolling mill bearing fault diagnosis test bench data is shown in Table 5. The accuracy of each model is lower than that of the AMVMD-MC1DCNN diagnostic model in this paper, which verifies the advantage of the model in this paper compared with existing models.

## 6. Conclusions

In this paper, we optimized the MVMD and 1DCNN algorithm models and proposed the AMVMD and MC1DCNN algorithm models to establish a fault diagnosis model for rolling mill multi-row roller bearings. Then, the DCGAN module is embedded in the model to improve the diagnostic accuracy of the model under unbalanced training set conditions. Additionally, the comparison with approximate and existing models verifies the advantages of the AMVMD-MC1DCNN model.

(1) We introduced GA to optimize the selection of important parameters K and α of MVMD, which improves the signal processing effects of MVMD. In addition, we introduced an iterative operator to accelerate the solution process of MVMD. The result comparison of AMVMD with MEMD and MVMD in processing the simulation signal showed that AMVMD could improve the signal processing speed, could effectively solve the parameter selection problem, and had a significantly better suppression effect on the modal mixing phenomenon than MVMD and MEMD.

(2) We introduced the same multichannel convolutional fusion layer in 1DCNN as MC1DCNN, which could make 1DCNN suitable for multi-channel signal processing. We combined both MC1DCNN and the 1DCNN with AMVMD and applied them to the rolling mill multi-row bearing fault diagnosis, and the correct rate of MC1DCNN was improved by 5.7% compared to 1DCNN input vertical vibration signals and by 4.2% compared to 1DCNN input axial vibration signals and by 2.6% compared to 1DCNN input mixed vibration signals.

(3) Under the conditions of three unbalanced ratio (1/5, 1/10 and 1/20) training sets, the accuracy of the fault diagnosis model after embedding the DCGAN module is improved by 12.5%, 17.0%, and 22.5%, respectively, compared with the original model.

The fault diagnosis model in this paper effectively achieves the identification of four faults of rolling mill multi-row bearings under unbalanced dataset conditions. The model has an important significance in the performance degradation assessment and multi-fault diagnosis of rolling mill multi-row bearings under unbalanced data conditions. 

Although the test stand largely simulates the actual working conditions of the rolling mill, the actual engineering signals are still very different from the experimental signals, and research work is still needed on how to further improve the effectiveness of signal processing in the current situation of deep learning for end-to-end fault diagnosis. Due to the instability of GAN in dataset enhancement, the model training is more difficult. However, existing research work on Wassertein GAN (WGAN) shows that the introduction of Wassertein distance in GAN solves both the problem of training instability and provides a reliable indicator of the training process. In this paper, we just used AMVMD to optimize the input of DCGAN and reduce the interference of invalid feature information to achieve the purpose of improving the performance of DCGAN. In the future, it is necessary for us to carry out work on improving the DCGAN network structure and improving its performance.

## Figures and Tables

**Figure 1 sensors-21-05494-f001:**
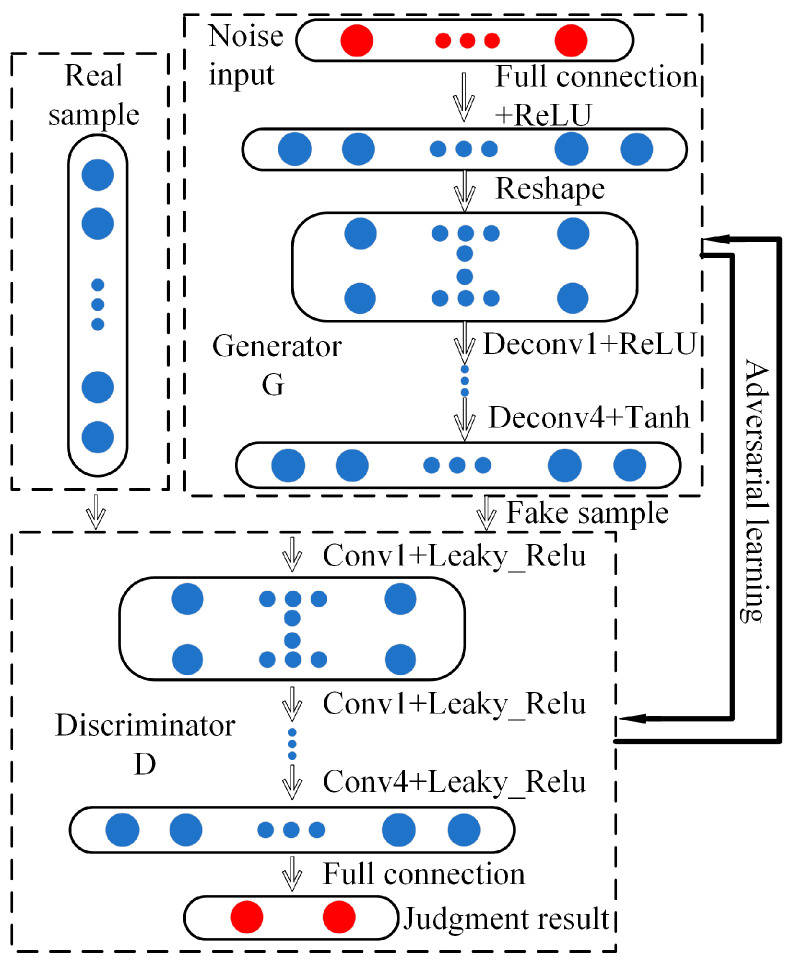
The structure of DCGAN.

**Figure 2 sensors-21-05494-f002:**
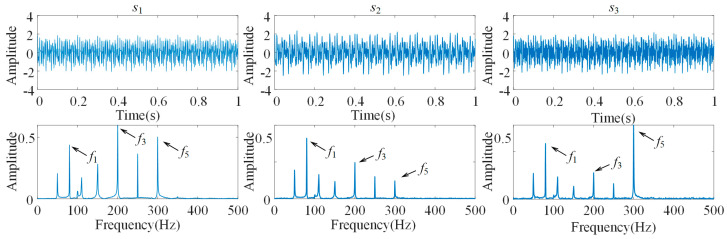
Time domain and frequency domain Figure of simulation signal.

**Figure 3 sensors-21-05494-f003:**
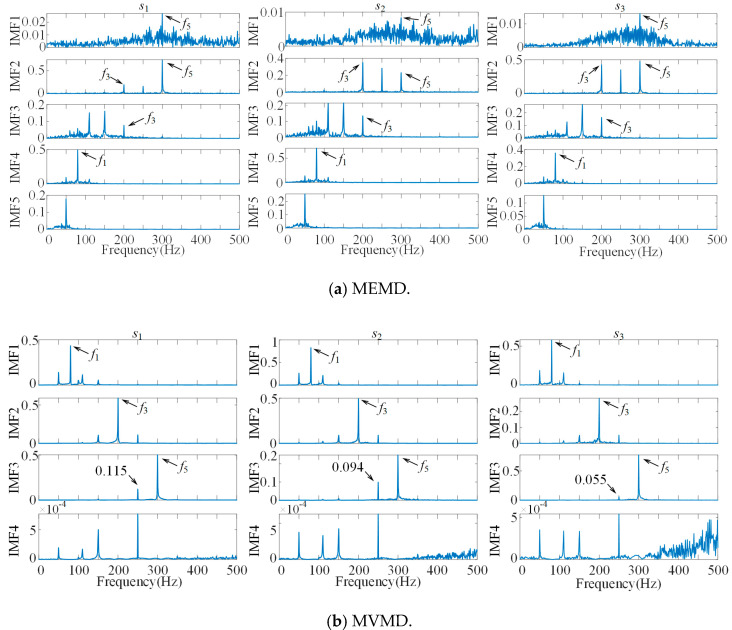
Effect comparison of three methods.

**Figure 4 sensors-21-05494-f004:**
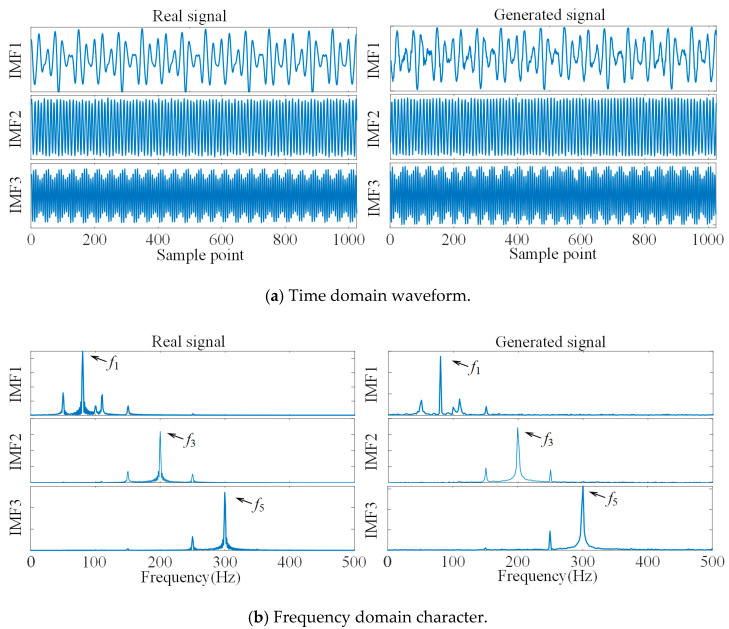
Comparison of real signal and generated signal.

**Figure 5 sensors-21-05494-f005:**
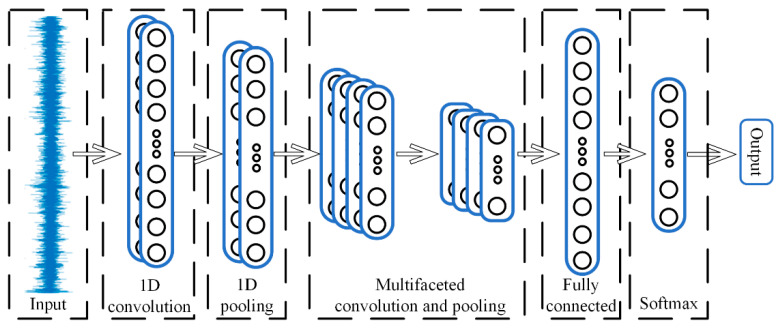
One-dimension convolutional neural network.

**Figure 6 sensors-21-05494-f006:**
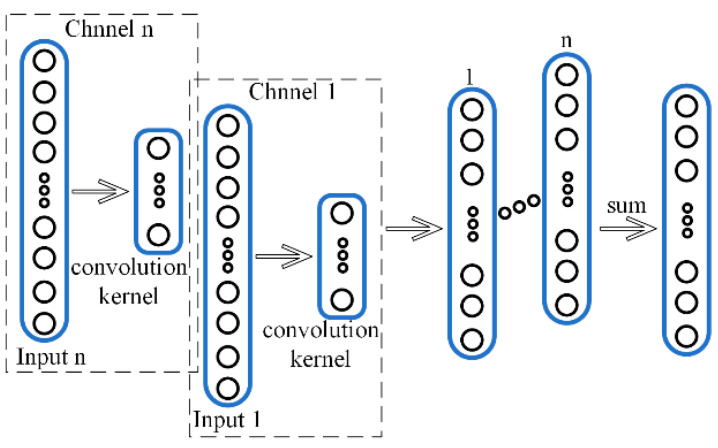
Multi-channel one-dimension convolutional fusion.

**Figure 7 sensors-21-05494-f007:**
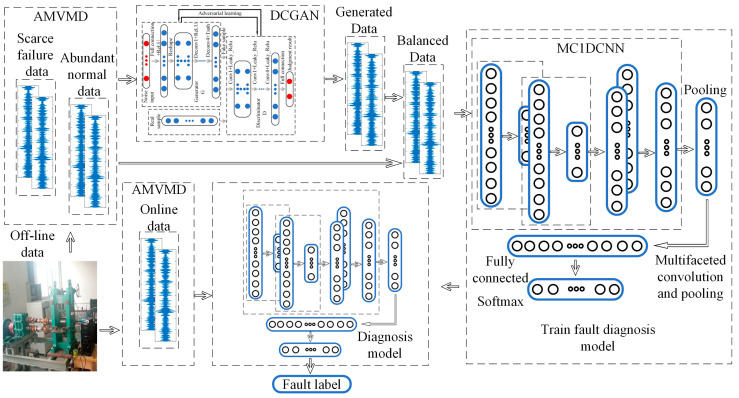
Fault Diagnosis Model.

**Figure 8 sensors-21-05494-f008:**
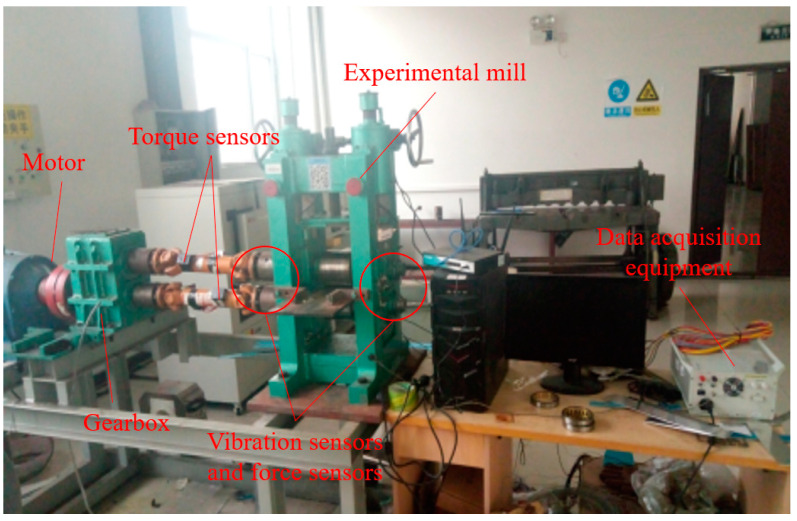
Experimental rolling mill bearing fault diagnosis test bench.

**Figure 9 sensors-21-05494-f009:**
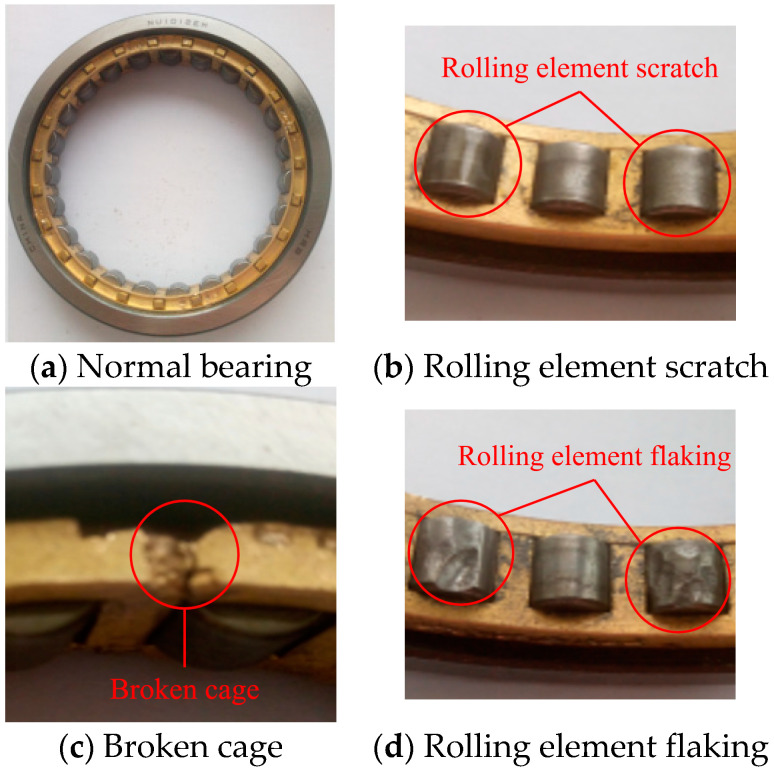
Four kinds of test bearing.

**Figure 10 sensors-21-05494-f010:**
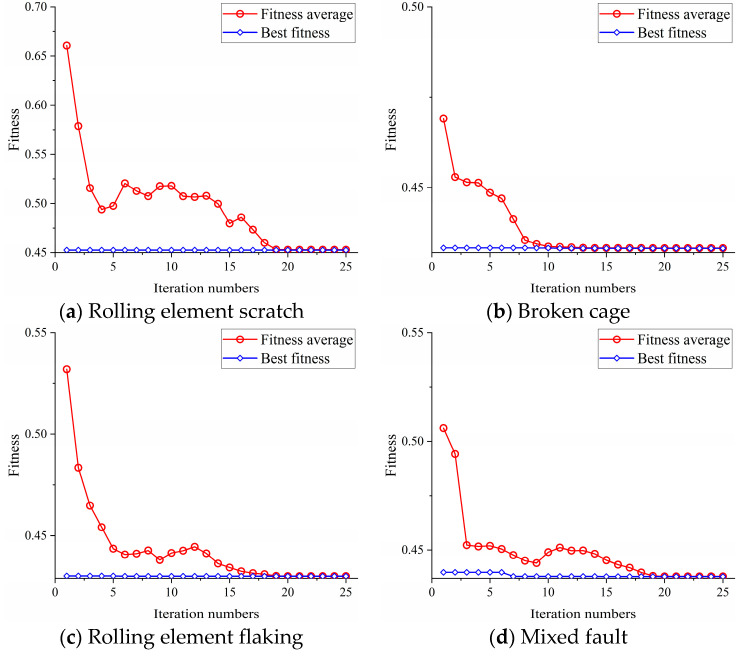
Optimization curve of GA.

**Figure 11 sensors-21-05494-f011:**
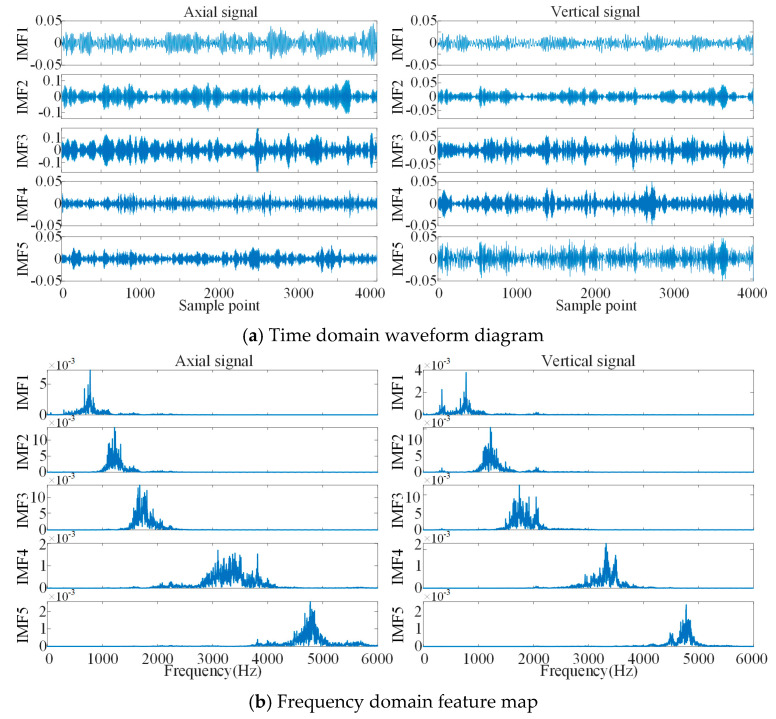
Decomposition results of bearings with rolling element flaking.

**Figure 12 sensors-21-05494-f012:**
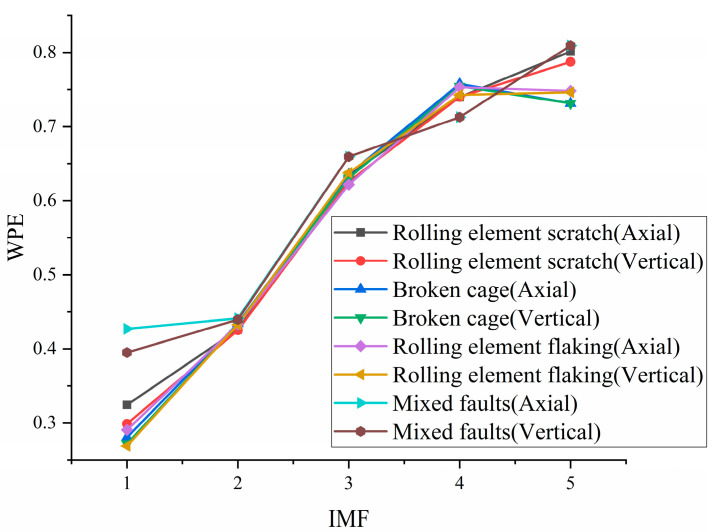
WPE of IMF of various signal.

**Figure 13 sensors-21-05494-f013:**
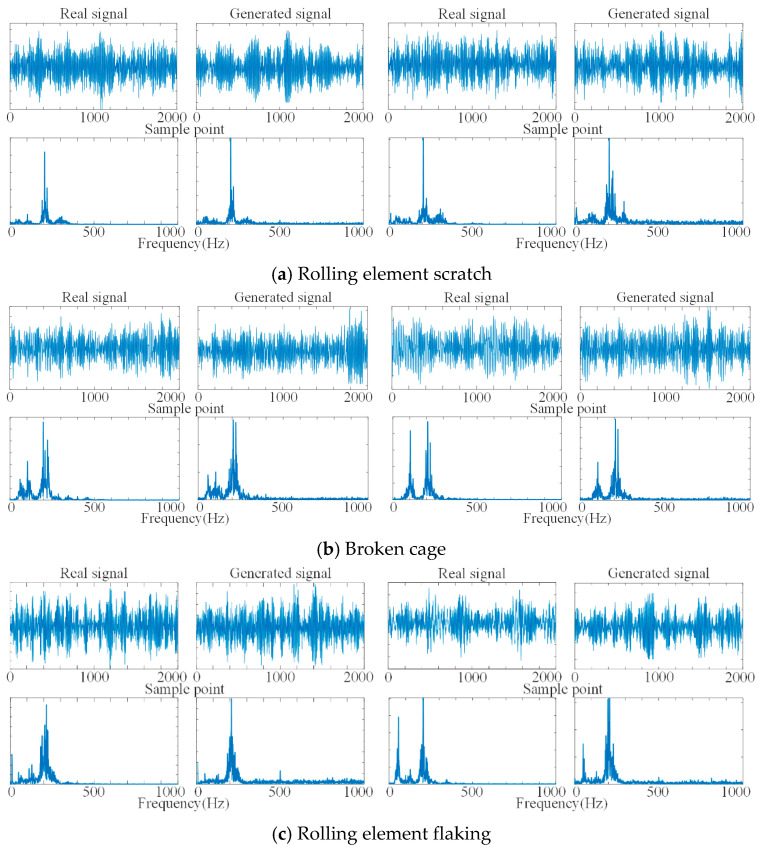
Comparison of time domain waveforms and frequency domain features of real and generated signals.

**Figure 14 sensors-21-05494-f014:**
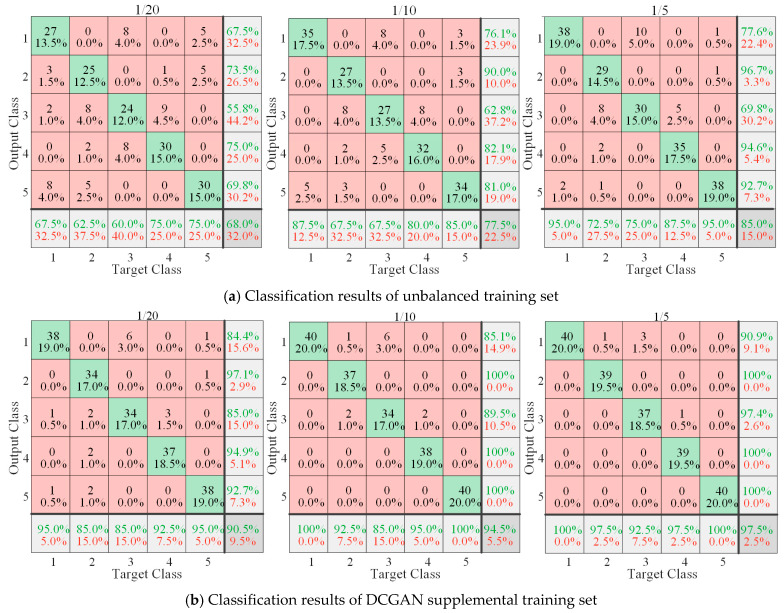
Confusion matrix for different diagnosis models.

**Figure 15 sensors-21-05494-f015:**
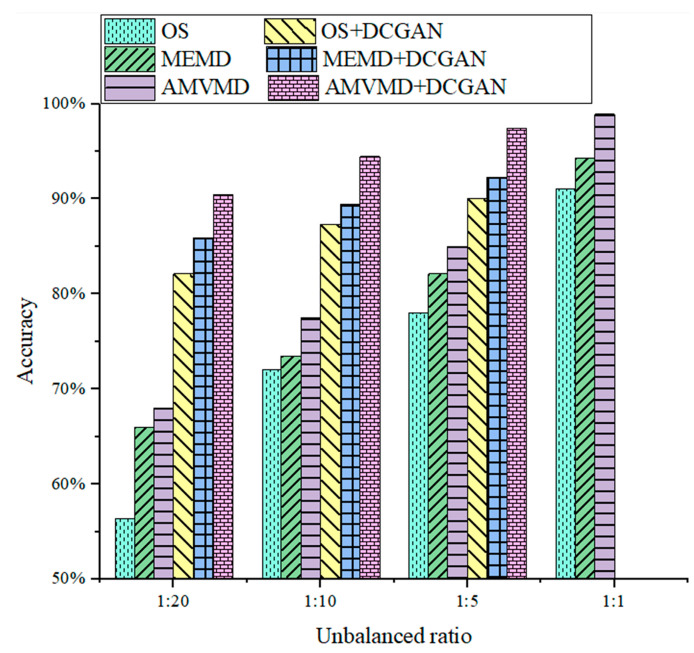
Diagnosis accuracy of various models under four scaled training sets.

**Table 1 sensors-21-05494-t001:** Comparison of operation time.

Length of Signal	Computation Time of MVMD(s)	Calculation Time of AMVMD (s)
2000	0.737	0.506
4000	3.345	2.001
6000	8.185	5.020
8000	11.887	7.241
10,000	15.817	9.642
12,000	22.567	14.037
14,000	30.682	19.367
16,000	41.052	26.477

**Table 2 sensors-21-05494-t002:** Optimized parameters for four types of fault signals.

Types of Faults	Number of IMF	Penalty Factor α
Rolling element scratch	5	2658
Broken cage	6	2941
Rolling element flaking	5	2931
Mixed faults	6	2137

**Table 3 sensors-21-05494-t003:** Network structure of MC1DCNN.

Network Structure	Convolution Kernel	Input Channel	Output Channel	Step	Activation Function
Convolutional layer 1	32 × 1	2	32	2	Tanh
Convolutional layer 2	4 × 1	32	64	2	ReLU
Convolutional layer 3	4 × 1	64	128	2	ReLU
Convolutional layer 4	4 × 1	128	128	2	ReLU

**Table 4 sensors-21-05494-t004:** Diagnosis accuracy of various fault diagnosis models.

Input Signal	Classification Model	Accuracy
Original signal (Vertical)	DBN	81.4%
Original signal (Axial)	DBN	84.2%
Original signal (Mixed)	DBN	89.8%
Original signal (Vertical)	1DCNN	84.3%
Original signal (Axial)	1DCNN	87.4%
Original signal (Mixed)	1DCNN	91.7%
MEMD reconstructed signal (Vertical)	DBN	87.3%
MEMD reconstructed signal (Axial)	DBN	89.5%
MEMD reconstructed signal (Mixed)	DBN	91.6%
MEMD reconstructed signal (Vertical)	1DCNN	87.5%
MEMD reconstructed signal (Axial)	1DCNN	90.1%
MEMD reconstructed signal (Mixed)	1DCNN	94.3%
AMVMD reconstructed signal (Vertical)	DBN	89.8%
AMVMD reconstructed signal (Axial)	DBN	91.3%
AMVMD reconstructed signal (Mixed)	DBN	95.4%
AMVMD reconstructed signal (Vertical)	1DCNN	93.2%
AMVMD reconstructed signal (Axial)	1DCNN	94.7%
AMVMD reconstructed signal (Mixed)	1DCNN	96.1%
AMVMD reconstructed signal	MC1DCNN	98.2%

**Table 5 sensors-21-05494-t005:** Diagnosis accuracy of existing fault diagnosis models.

Vibration Signal	Classification Model	Model Input	Accuracy
Vertical signal	VMD-ELM	Multidomain features	90.4%
Axial signal	VMD-ELM	Multidomain features	92.6%
Mixed signal	VMD-ELM	Multidomain features	94.5%
Vertical signal	MVMD-SVM	MWPE	90.2%
Axial signal	MVMD-SVM	MWPE	91.4%
Mixed signal	MVMD-SVM	MWPE	94.7%
Vertical signal	VMD-CNN	The reconstructed signal	92.3%
Axial signal	VMD-CNN	The reconstructed signal	93.4%
Mixed signal	VMD-CNN	The reconstructed signal	95.1%

## Data Availability

The experimental data can be obtained by asking at 136145855@qq.com. The experimental data is obtained through the rolling experiment of National Cold Rolling Strip Equipment and Process Engineering Technology Research Center of Yanshan University. The experimental results are reproducible. Relevant scholars can use similar experimental models or go to the National Cold Rolling Strip Equipment and Process Engineering Technology Research Center of Yanshan University to further verify the reliability of the experimental data.

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
