# Peer review of "Fault Diagnosis Method for Rolling Mill Multi Row Bearings Based on AMVMD-MC1DCNN under Unbalanced Dataset"

_sensors, 2021, doi:10.3390/s21165494_

Round 1

Reviewer 1 Report

An interesting paper that combines AMVMD with MC1DCNN in order to detect rolling mills multi-row rolling bearing faults, using multi-channel data analysis. DCGAN is embedded in the model to solve the problem of available unbalanced training datasets.

The proposed fault diagnosis system has been tested using data acquired from a test bench for 4 fault types: rolling element scratch; broken cage; rolling element flaking and mixed faults. The proposed method provides very good accuracy compared to other similar methods.

Regarding the desricpoin of the mathematical models that has been used an implemented in the proposed fault diagnosis system author should check equation 11 (GA crossover model equation) it should be X'1=λX1+(1-λ)X2 instead of X'1=λX1+(1-λ)X1

Author Response

1. Response to comment: (An interesting paper that combines AMVMD with MC1DCNN in order to detect rolling mills multi-row rolling bearing faults, using multi-channel data analysis. DCGAN is embedded in the model to solve the problem of available unbalanced training datasets.)

    Response: We are very grateful to the reviewer for the comment and recognition of this paper.

2. Response to comment: (The proposed fault diagnosis system has been tested using data acquired from a test bench for 4 fault types: rolling element scratch; broken cage; rolling element flaking and mixed faults. The proposed method provides very good accuracy compared to other similar methods.)                              

    Response: Once again, we are very grateful to the reviewer for the comment and recognition of this paper.

3. Response to comment: (Regarding the description of the mathematical models that has been used an implemented in the proposed fault diagnosis system author should check equation 11 (GA crossover model equation) it should be X'1=λX1+(1-λ)X2instead of X'1=λX1+(1-λ)X1                                                                       

    Response: We specially thank the reviewer for the careful review, and we have revised equation (11) as followings:

         X'1=λX1+(1-λ)X2

         X'2=λX2+(1-λ)X1

Author Response

1. Response to comment: (Paper is well-written and interesting; Content is original, presentation is clear and the research can contribute to bringing more knowledge into the field of research;)
    Response: We are very grateful to the reviewer for the comment and recognition of this paper.

2. Response to comment: (English language and style are minor spell check required)                               

    Response: We are very grateful to the reviewer for careful reading of this  paper. We have checked this article for     spelling issues, and all changes can be viewed in revision mode.

3. Response to comment: (Please add the author initials and their e-mails in the authors section)                

    Response: We have added the author initials and their e-mails in the authors’ section.

4. Response to comment: (Please check the spacing before and after the Figures and Tables, in order to meet the requirements of the journal template. Same recommendation for the subsection titles.)                                   

    Response: We have made changes to the formatting of images, tables and subsection title. And ensure that images and tables are spaced 12 pounds apart from the preceding and following text, ensure the headings are spaced 12 pounds apart from the preceding text and 3 pounds apart from the following text.

5. Response to comment: (Please enlarge Figure 14 in order to ensure a better readability)                          

    Response: We have modified the size of image 14 to ensure a better readability.

6. Response to comment: (Please eliminate Section 7 as there are no patents resulting from the work reported in this manuscript.)                                                                                                                                           

    Response: We have eliminated Section 7.

7. Response to comment: (Please review the reference list in order to meet the requirements of the journal template.)                                                                                                                                                             

    Response: We have made changes to the format of the references and added the DOI URL of the reference.

8. Response to comment: (The literature review from the Introduction section is poor. Please add some more actual references)                                                                                                                                                   

    Response: It is really true as Reviewer suggested that the literature review from the Introduction section is poor. We have made more changes to the literature review and added some references. We have removed some conference papers and re-cited some references in related fields and re-discussed the authors' work. For the relevant areas where references were missing from the review, we added some actual references and further summarized the work of scholars. All changes are made in revision mode and can be viewed in the revised version. We are sorry to say that we did not find some related literature on the MVMD algorithm due to the small number of applications.

9. Response to comment: (In Section1, the main contributions of the paper should be better explained)  

    Response: We have modified this part according to the Reviewer’s suggestion and added the section structure part of the article. We have cited relevant literature to illustrate the shortcomings of existing methods to further highlight the contribution of our approach. Modified content can be viewed in revision mode.

10. Response to comment: (There is lack of comparison with the literature. In particular, it is essential that the authors demonstrate the quality or efficiency of their results, compared to well-established methods)     

    Response: It is really true as Reviewer suggested that we need to add comparison with the literature and we have added this part in the end of the paper. The models selected for comparison are VMD-ELM model of literature [15], MVMD-SVM model of literature [17] and VMD-CNN model of literature [22]. These models are chosen because of their closer approximation to the models in this paper and the large amount of existing research that has demonstrated the advantages of VMD in modal decomposition algorithms and CNN in neural network classifiers, so methods such as EMD or EWT and networks such as BP or DNN are not added to the comparison. Since both the VMD-ELM model and the MVMD-SVM model require feature extraction of the vibration signal, in the process of performing MWPE feature extraction, we found that the larger embedding dimension and scale factor of the MWPE algorithm increase the computing time significantly, so the CNN model has an absolute advantage in diagnostic time after training is completed. Therefore, we ignore the time required for feature extraction and compare only the diagnostic accuracy of the models. A more comprehensive comparison between 1DCNN and MC1DCNN has been made in the results of this paper, so the 1DCNN model of the literature [25] and [26] is not compared subsequently. We have reproduced the mentioned model as far as possible and used it in the analysis of the experimental rolling mill bearing fault diagnosis test bench data in this paper. The analysis results are compared with the model in this paper and the comparison further validates the advantages of the model in this paper. We have added this section to the end of the article. All changes are available for viewing in revision mode.

11. Response to comment: (The Conclusion section is sooner a summary of the work, without really presenting the conclusions that can be drawn from this research. Therefore, the quantitative results are required, and the meaningfulness of this study would be emphasized rather than presenting a summary on the technical works. Furthermore, the section should be written more comprehensively and it has to be extended with the weak points of the proposed method and further studies.)                                                                                                       

    Response: We have re-written this part according to the Reviewer’s suggestion. We have added a quantitative comparison to the conclusions of the algorithm in this paper, such as the accuracy improvement values under different conditions. We have further summarized the effect of the algorithm in this paper on the performance improvement of fault diagnosis models compared with existing methods and highlighted the significance of this paper's research even more. Finally, we propose some shortcomings in the research method and future work that can be carried out in this paper to address the shortcomings in signal processing and the performance of DCGAN in the fault diagnosis process as follows:Although the test stand largely simulates the actual working conditions of the rolling mill, the actual engineering signals are still very different from the experimental signals, and research work is still needed on how to further improve the effectiveness of signal processing in the current situation of deep learning for end-to-end fault diagnosis. Due to the instability of GAN in dataset enhancement, the model training is more difficult. However, existing research work on Wassertein GAN (WGAN) shows that the introduction of Wassertein distance in GAN solves both the problem of training instability and provides a reliable indicator of the training process. In this paper, we just used AMVMD to optimize the input of DCGAN and reduce the interference of invalid feature information to achieve the purpose of improving the performance of DCGAN. In the future, it is necessary for us to carry out work on improving the DCGAN network structure and improving its performance.

Special thanks to you for your good comments.

Reviewer 3 Report

No serious objections. DCGAN is an interesting way to address the lack of fault data. 

Very well structured and presented work. 

Author Response

1. Response to comment: (No serious objections. DCGAN is an interesting way to address the lack of fault data. Very well structured and presented work.)
    Response: We are very grateful to the reviewer for the comment and recognition of this paper.